# Species-Level Classification of Peatland Vegetation Using Ultra-High-Resolution UAV Imagery

Gillian Simpson [1,2,*], Caroline J. Nichol [1], Tom Wade [1], Carole Helfter [2], Alistair Hamilton [3] and Simon Gibson-Poole [3]

1   School of Geosciences, University of Edinburgh, Alexander Crum Brown Road, Edinburgh EH9 3FF, UK; caroline.nichol@ed.ac.uk (C.J.N.); tom.wade@ed.ac.uk (T.W.)
2   UK Centre for Ecology & Hydrology (UKCEH), Bush Estate, Penicuik EH26 0QB, UK; caro2@ceh.ac.uk
3   Scotland's Rural College (SRUC), Peter Wilson Building, King's Buildings, West Mains Road, Edinburgh EH9 3JG, UK; alistair.hamilton@sruc.ac.uk (A.H.); simon.gibson-poole@sruc.ac.uk (S.G.-P.)
*   Correspondence: gillian.simpson@slu.se

**Abstract:** Peatland restoration projects are being employed worldwide as a form of climate change mitigation due to their potential for long-term carbon sequestration. Monitoring these environments (e.g., cover of keystone species) is therefore essential to evaluate success. However, existing studies have rarely examined peatland vegetation at fine scales due to its strong spatial heterogeneity and seasonal canopy development. The present study collected centimetre-scale multispectral Uncrewed Aerial Vehicle (UAV) imagery with a Parrot Sequoia camera (2.8 cm resolution; Parrot Drones SAS, Paris, France) in a temperate peatland over a complete growing season. Supervised classification algorithms were used to map the vegetation at the single-species level, and the Maximum Likelihood classifier was found to perform best at the site level (69% overall accuracy). The classification accuracy increased with the spatial resolution of the input data, and a large reduction in accuracy was observed when employing imagery of >11 cm resolution. Finally, the most accurate classifications were produced using imagery collected during the peak (July–August) or early growing season (start of May). These findings suggest that despite the strong heterogeneity of peatlands, these environments can be mapped at the species level using UAVs. Such an approach would benefit studies estimating peatland carbon emissions or using the cover of keystone species to evaluate restoration projects.

**Keywords:** UAV; multispectral imagery; peatland; supervised classification; single species; Parrot Sequoia; centimetre-scale resolution

## 1. Introduction

### 1.1. Peatland Environments and Their Significance

Northern peatlands (>45° N) are historical carbon sinks of global importance, estimated to store twice as much carbon as the world's forest biomass [1,2]. The cool, wet climate experienced by these ecosystems maintains a high water table and inhibits the decomposition of organic matter [3,4], resulting in the accumulation of carbon in the form of peat. At present, many of these ecosystems are severely degraded, having been drained for agriculture or forestry over the past millennium. The exploitation of these environments has released significant quantities of historical carbon into the atmosphere and negatively impacted ecosystem functioning [5]. Over the past decade, however, peatlands have become increasingly recognised in climate change mitigation, with the protection and restoration of these ecosystems now forming a key part of many climate agreements [6–9]. In terms of restoration projects, a number of indicators, such as the water-table depth and the presence and cover of keystone species such as *Eriophorum* (cotton-grass) and *sphagnum* mosses, can be used to evaluate project success [10–12]. Furthermore, as plant functional types (PFTs; e.g., shrubs, bryophytes, graminoids) are known to influence ecological processes, including emissions of carbon dioxide and methane [13–15], vegetation surveys and

maps can be used to estimate carbon emissions [16]. These two approaches demonstrate the large potential for UAV-based monitoring of peatlands and their vegetation.

### 1.2. Monitoring Peatland Vegetation

Despite the global significance of peatlands, the monitoring of these ecosystems has been rather limited. Peatland environments are often inaccessible, being located in geographically remote areas. In addition, their 'boggy' nature means that peatlands can be impassible and require extensive boardwalks for frequent monitoring. As a result, detailed ground-based botanical surveys are rare and expensive to conduct. Vegetation mapping has instead often been conducted by remote sensing, via the use of satellite and airborne imagery [17–19]. Advancements in sensor technology have made monitoring possible at increasing spectral and spatial resolutions. Satellites such as QuickBird, IKONOS, and PlanetScope, for example, can view peatlands at spatial scales in the order of metres or less [20–22]. However, the frequent presence of cloud cover in boreal and temperate regions (where many peatlands are located) makes it difficult to conduct frequent vegetation mapping based on optical satellite data (e.g., [23,24]). The development of Uncrewed Aerial Vehicle (UAV) platforms and innovations in sensor technology have therefore been key for the remote sensing of peatlands. As such, UAVs now offer an affordable, repeatable and flexible method for examining peatland vegetation at ultra-high (centimetre-scale) resolution [25,26].

Despite these advancements in the spatial resolution of data, much work has focused on the mapping of distinct habitat types at the landscape level. Anderson et al., for example, mapped peatland biotopes across a raised bog in the UK using IKONOS data (4 m resolution), with classes ranging from 'active raised bog' to 'drained' and 'degraded bog' [20]. Similarly, Palace et al. used 3 cm resolution UAV data to classify broad vegetation cover (e.g., 'hummock', 'wet', and 'tall shrub') in northern Sweden [27]. At a smaller scale, research has been conducted to map individual PFTs (e.g., [28,29]). Although changes in the distribution of PFTs over time may be useful for indicating long-term changes in carbon storage for example [30,31], PFT-level classifications mask a large amount of heterogeneity. Few studies have examined the feasibility of species-level classifications, and those that have done often focused on a single species of interest only [29,32,33].

### 1.3. Challenges for Remote Sensing in Peatland Environments

Peatland environments are complex and display variability across spatial scales. Hence, despite appearing homogeneous at the landscape scale ($100$–$10^6$ $m^2$), at the microsite scale ($0.1$–$1$ $m^2$) they display strong heterogeneity. This complexity makes peatlands challenging environments for remote sensing [26], particularly when mapping at fine spatial scales. Firstly, peatland vegetation is characterised by a multi-layer canopy, which makes it problematic to sense from above [34,35]. Secondly, the characteristic hummock–hollow terrain in these environments creates a topographic shadow in remote sensing imagery. This shadow affects the spectral reflectance measured by sensors and can reduce the classification accuracy [36–38]. Finally, peatlands are not static environments. Many vegetation species display a clear phenological cycle over the growing season: greening during the spring, flowering, and then senescing in the autumn. This phenology can have a strong impact on the spectral signatures of individual species [39] and poses a challenge for accurate mapping if using single time frames.

### 1.4. Study Overview and Objectives

This study examines the use of ultra-high-resolution UAV imagery to classify vegetation in a temperate peatland. For this purpose, we collected true-colour (RGB) and multispectral imagery at monthly intervals over a complete growing season. With regard to the current literature, ours appears to be one of the highest spatial and temporal resolution datasets available for a peatland environment, and one of only a handful of remote-sensing studies mapping peatland vegetation at the species level [29,40,41]. The objectives of this

study are: (1) to explore the accuracy of supervised techniques for classifying vegetation at the species level; and (2) to examine the impact of the temporal sampling date and spatial resolution on the classification accuracy.

## 2. Materials and Methods

### 2.1. Study Area

This study examines the vegetation at Auchencorth Moss, a low-lying ombrotrophic peatland located in south-east Scotland, UK (55°47′33 N, 3°14′37 W; 267 m a.s.l.). The peatland covers an area of around 10 km$^2$ and was historically affected by drainage. Over time, the site has naturally rewetted and the old drainage ditches are now largely overgrown. The peatland is currently used for low-density sheep grazing (~1 per ha) and hosts a long-term atmospheric observatory and monitoring site for greenhouse gas emissions [42,43].

Blanket bog characterises the site, which exhibits a strong hummock–hollow micro-topography typical of many peatlands. The hummocks are around 40 cm in diameter and 30 cm tall. These raised features are dominated by vascular plants, which form the upper canopy, consisting of sedges (e.g., *Eriophorum vaginatum*), rushes (e.g., *Juncus effusus*) and grasses (e.g., *Deschampsia flexuosa*), and are underlain by a carpet of mosses (e.g., *Polytrichum commune*, *Sphagnum* spp.). The hollows, in contrast, are characterised by predominantly wet conditions due to their lower elevation. These features have a lower coverage of vascular plants and are often dominated by mosses.

### 2.2. UAV Data Collection and Processing

UAV data were collected at Auchencorth Moss at least monthly from May to October 2021, covering an area of around 0.05 km$^2$. Before the growing season commenced, a series of markers were placed within the survey area. These included ten Ground Control Points (GCPs) to aid in geo-referencing the acquired imagery during processing, and five Check Points (CPs) to assess the geolocation accuracy of the produced maps. All the markers were surveyed using a high-precision Global Navigation Satellite System (GNSS; GPS500, Leica Geosystems AG, Heerbrugg, Switzerland) to compute their coordinates with a horizontal accuracy of <2 cm.

Two types of UAV data were collected: (1) RGB data using a Mavic Pro 2 (SZ DJI Technology Co., Ltd., Shenzhen, China); and (2) multispectral data using a Parrot Sequoia sensor (Parrot Drones SAS, Paris, France). More detail on each type of data collected is provided in Table 1.

**Table 1.** Overview of the types of UAV data collected at Auchencorth Moss during the 2021 growing season. Ground sampling distance (GSD) describes the physical distance between adjacent pixel centres in the imagery, i.e., for a 2 cm GSD each pixel corresponds to a 2 cm distance on the ground. The accuracy of each dataset is indicated by the calculated root mean square error (RMSE) values. Hereby, the observed locations of five Check Points (CPs) in the processed imagery were compared against their corresponding ground-surveyed GNSS (Global Navigation Satellite System) coordinates. Note that these CPs were not used during image processing and hence provide an independent assessment of the geolocation accuracy in the processed imagery. For the multispectral data, the GSD and RMSE represent mean values (averaged over all flights during the growing season); for the RGB imagery, data were employed from a single survey conducted on 18 May 2021.

| Sensor | Data Type | Survey Height (m) | Image Overlap (% Front, Side) | GSD (cm) | RMSE [x, y, z] (cm) |
|--------|-----------|-------------------|-------------------------------|----------|---------------------|
| **Mavic Pro 2** | RGB | 65 | 70, 80 | 1.46 | 1.12<br>0.97<br>4.07 |
| **Parrot Sequoia** | Multispectral | 25 | 80, 80 | 2.80 | 1.35<br>1.47<br>6.50 |

Data collection was restricted to dates with low average wind speed ($<5$ m s$^{-1}$), which were either consistently overcast or clear to minimise variation in the scene illumination during image acquisition. The meteorological conditions were logged on the flight dates and all the UAV surveys were conducted within 2 h of solar noon to reduce the impact of shadow on the imagery collected. Both the RGB and multispectral flights surveyed the same ground area. For the latter, the Mission Planner software (v1.3.5, ArduPilot, http://www.ardupilot.org, accessed on 15 October 2017) was used to design an autonomous flight plan for repeat surveys to ensure the comparability of the collected data throughout the growing season. The Mavic 2 Pro RGB missions were planned and flown using the DJI Ground Station Pro app for iOS (https://www.dji.com/se/ground-station-pro, accessed on 24 January 2019).

### 2.2.1. Multispectral Data

Multispectral data were collected by a Parrot Sequoia sensor mounted on a Tarot T680 Pro frame (Wenzhou Tarot Aviation Technology Co., Wenzhou, China). This custom-built hexacopter ran the open-source Arducopter autopilot firmware on a PixHawk flight controller, and it is shown in Figure 1 alongside photographs of the survey area. The Parrot Sequoia sensor consists of two components. The first is a downward-facing multispectral camera, which measures spectral reflectance in the green (530–570 nm), red (640–680 nm), red-edge (730–740 nm), and near-infrared (NIR; 770–810 nm) bands. This camera was mounted on a Tarot 2D gimbal to provide stabilisation of the pitch and roll axes. The second component is the 'sunshine' or irradiance sensor, which was installed on a fixed mount on the top surface of the UAV above the level of the aircraft's GNSS antennae, with a clear sky view. This upward-facing incident light sensor records changes in incoming irradiance over the period of image capture and automatically corrects the measured reflectance from the downward-facing multispectral camera.

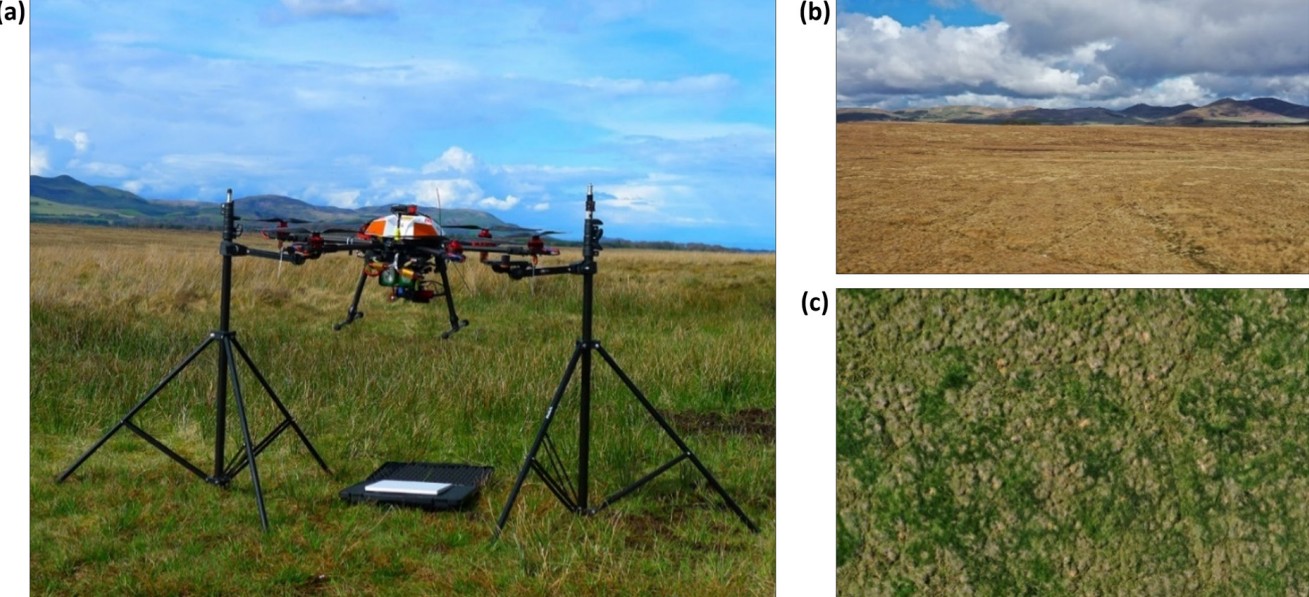

**Figure 1.** UAV surveying at Auchencorth Moss. Shown in (**a**) is the custom-built Tarot T680 Pro (Wenzhou Tarot Aviation Technology Co., Wenzhou, China) and mounted Parrot Sequoia camera positioned over a spectral calibration panel (photo credit: G. Simpson). Shown on the right are photographs taken with the Mavic 2 Pro (SZ DJI Technology Co., Ltd., Shenzhen, China): (**b**) is an aerial photograph of the survey area; and (**c**) shows one of the individual images acquired during the RGB survey, which highlights the strong spatial heterogeneity of the site.

The four bands sampled by the multispectral camera are well-suited for detecting the characteristic spectral features of vegetation, including a reflectance peak in the green band

and a well-known chlorophyll absorption feature in the red band [44,45]. The red-edge and NIR bands in particular are known to be important for discriminating between peatland vegetation communities (e.g., [46]). The reflectance in the red-edge, for example, is related to the chlorophyll content and leaf area [47–49], whereas the reflectance in the NIR band is a good indicator of the leaf water content [50]. Multispectral surveys were conducted on seven days during the 2021 growing season: 14 May, 2 June, 23 July, 3 August, 25 August, 20 September, and 15 October. An overview of the solar conditions on each survey day is provided in Figure 2.

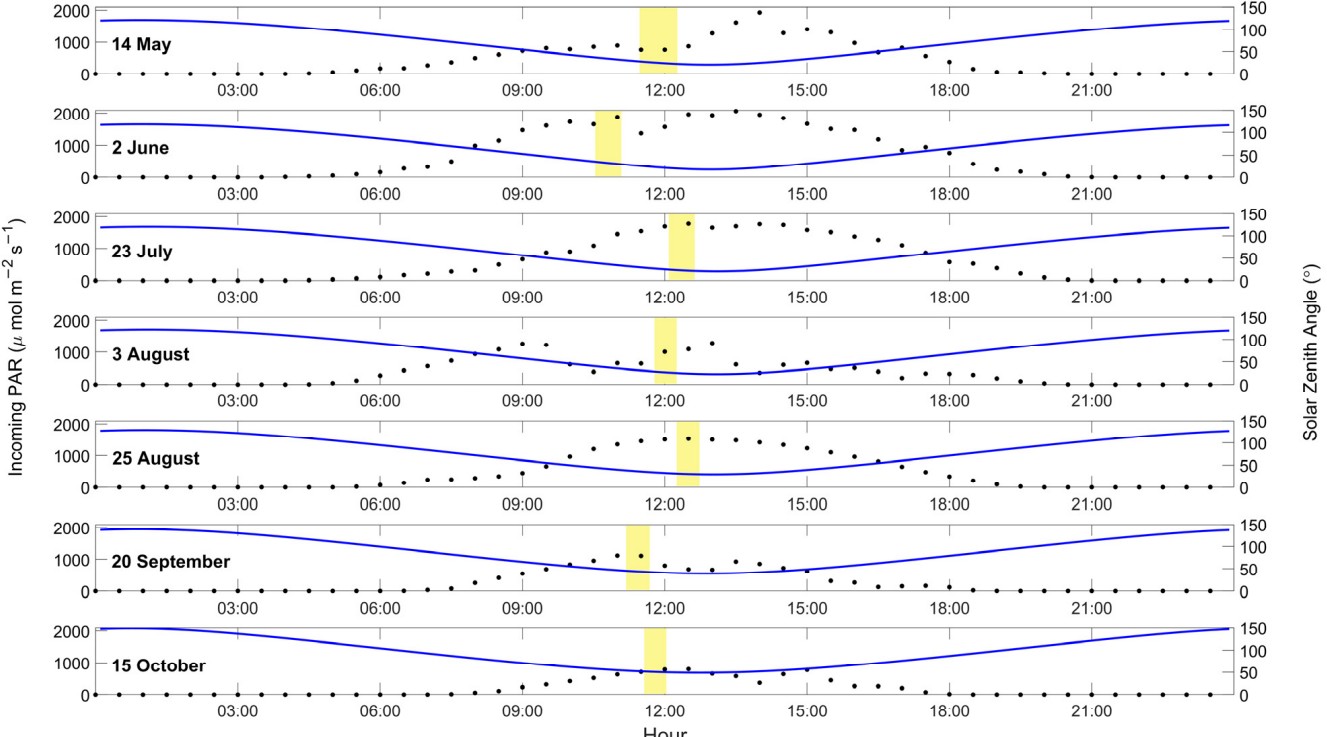

**Figure 2.** Diurnal overview of the solar conditions on the multispectral survey days. Shown are: the period of UAV image acquisition (yellow shaded areas); half-hourly incoming photosynthetically active radiation (PAR) measured at the site (black dots; SKP215, Skye Instruments Ltd., Llandrindod Wells, UK), and solar zenith angle (blue line) calculated using the NOAA Solar Calculator (https://gml.noaa.gov/grad/solcalc/, accessed on 21 January 2022). From top to bottom are the survey days: 14 May (overcast), 2 June (clear), 23 July (clear), 3 August (overcast), 25 August (clear), 20 September (clear), and 15 October 2021 (clear).

The acquired multispectral imagery was processed using the 'Ag multispectral' template in the photogrammetry software Pix4DMapper (v4.4.12, Pix4D S.A., Prilly, Switzerland). The initial processing produced a rough model from the imagery based on the identification and matching of key points. This model was then geo-referenced (aligned to a geographic coordinate system) through the marking of ten GCPs to improve estimation of the camera positions and lens parameters. As a next step, radiometric correction of the imagery was conducted using images from a calibrated reference panel (50% reflectance, SphereOptics GmbH, Herrsching am Ammersee, Germany) taken during each flight, before a separate reflectance map was produced for each of the four bands. Finally, shadow was removed from the processed imagery prior to classification. For this purpose, a new green–red band (GR$_{av}$) was created as follows:

$$GR_{av} = \frac{\rho_{red} + \rho_{green}}{2}$$

where $\rho$ is the measured reflectance in the red and green bands, respectively. For each survey date, 'shadow' was defined as all the pixels with $GR_{av}$ less than 1 standard deviation below the mean, and it was masked from all the vegetation classifications.

### 2.2.2. RGB Data

The RGB imagery was collected on 3 August 2021 using the Mavic Pro 2 and processed using the Pix4D '3D Maps' template. Hereby, the Structure from Motion (SfM) technique was used to generate an RGB orthomosaic and top-of-canopy Digital Surface Model (DSM) of the survey area. Again, the ten surveyed GCPs were used to geo-reference the model, and an overview of its accuracy is provided in Table 1.

The DSM highlighted the strong micro-topography at Auchencorth Moss and the small E–W elevation gradient (5 m) over the surveyed area. As such, and in preparation for the classification analysis, two additional elevation datasets were created so that the pixels could be compared across the image at random. The first of these was a detrended elevation map, created using a low-pass filter to smooth out any micro-topography. The second was a normalised DSM, calculated as the difference between the detrended elevation map and the original DSM. This normalised dataset allowed for the differentiation of hummocks (local high-points) and hollows (local low-points), regardless of the pixel location.

### 2.3. Ground Validation Data

The ground validation data provided an independent measure for accuracy assessment and allowed for the training and validation of classifications based on the UAV data. Two types of ground validation data were employed in the present study based on: (i) ground-level spectral reflectance measurements of individual species in the field; and (ii) high-accuracy GNSS measurements of dominant vegetation patches within the UAV survey area. GNSS point data were employed for ground validation across all the UAV survey dates. In contrast, the use of spectral reflectance measurements as ground validation data was restricted to two instances when UAV surveying was conducted within 1 week of the ground spectral measurements. The classification analyses presented in the Results section (Section 3) therefore employ GNSS point data as ground validation unless stated otherwise.

### 2.3.1. Spectral Reflectance Measurements

An ASD field spectroradiometer (FieldSpec Pro, Analytical Spectral Devices, Inc., Longmont, CO, USA) was used to measure the reflectance of vegetation at Auchencorth Moss in the range 350–2500 nm on 2 June and 23 July 2021. Both dates were characterised by clear skies and low wind speeds (<5 m s$^{-1}$), and (as with the UAV surveys) the ASD measurements were conducted within two hours of solar noon. The ASD was operated in 'White Reference' mode, whereby the reflectance of a target was calculated relative to a calibrated white reference standard (Spectralon target; Labsphere, Inc., North Sutton, NH, USA) to allow the spectral measurements to be visualised in real time. Hand-held measurements were performed using the bare fibre in a mounting block (23° field of view), which was positioned close to the nadir ca. 30 cm above each target, corresponding to a field of view on the ground of around 12 cm. Each measurement consisted of 50 scans averaged by the spectrometer to minimise the signal-to-noise ratio in the collected spectra. Finally, to facilitate comparison with the multispectral UAV data, spectra in the range of the four Parrot Sequoia bands were extracted from the processed ASD spectra.

### 2.3.2. GNSS Measurements

Due to the patchy and heterogeneous nature of the vegetation at Auchencorth Moss and despite expert knowledge of the site, it was not possible to identify individual species based on visual inspection of the UAV imagery alone. To remedy this, a detailed field survey of vegetation across the UAV survey area was conducted over four days from October–November 2021. The purpose of this survey was to provide areas of spatial training data for classification and validation data against which to assess accuracy. These

GNSS point data were employed throughout the study period. It was hence assumed that the spatial location of individual vegetation species in the peatland was not affected by their phenological cycle.

Restricting the vegetation survey to the upper canopy (i.e., the portion of vegetation observed in the UAV imagery), eleven dominant species were identified in the UAV survey area. These are depicted in Figure 3 and span multiple PFTs, from mosses (*Pleurozium schreberi*, *Sphagnum* spp., *Polytrichum commune*) to sedges (*Eriophorum vaginatum*), rushes (*Juncus effusus*), grasses (*Molinia caerulea*, *Deschampsia flexuosa*), forbs (*Potentilla erecta*) and shrubs (*Vaccinium* spp., *Calluna vulgaris*, *Erica tetralix*). Although it was uncommon to find 'pure' patches of vegetation for each species, with grasses often emerging through shrub patches for example, many patches were dominated by a single species (defined as average upper canopy cover >80%). The point coordinates around these patches were surveyed to create single-species polygons (i.e., regions of interest, ROIs). The ROIs comprised a total of 256 polygons spanning the eleven species (Table 2). Species such as *J. effusus* often formed extensive patches (>25 m$^2$), whereas others were less abundant, with a much smaller patch size (e.g., *E. tetralix*, *P. erecta*; ca. 0.01–0.02 m$^2$). As such, although this 'on the go' sampling design achieved relatively good coverage of species over the UAV survey area, time constraints and the heterogeneous nature of the vegetation at Auchencorth Moss meant it was not possible to collect the same number or size of ROIs for each vegetation species (see Table 2). These single-species ROIs were divided into training (~70%) and validation datasets (~30%), each containing a similar ratio of image pixels and polygons for each single-species ROI. For each flight day, the spectral properties of the ROIs were extracted by overlaying their coordinates on the produced reflectance maps.

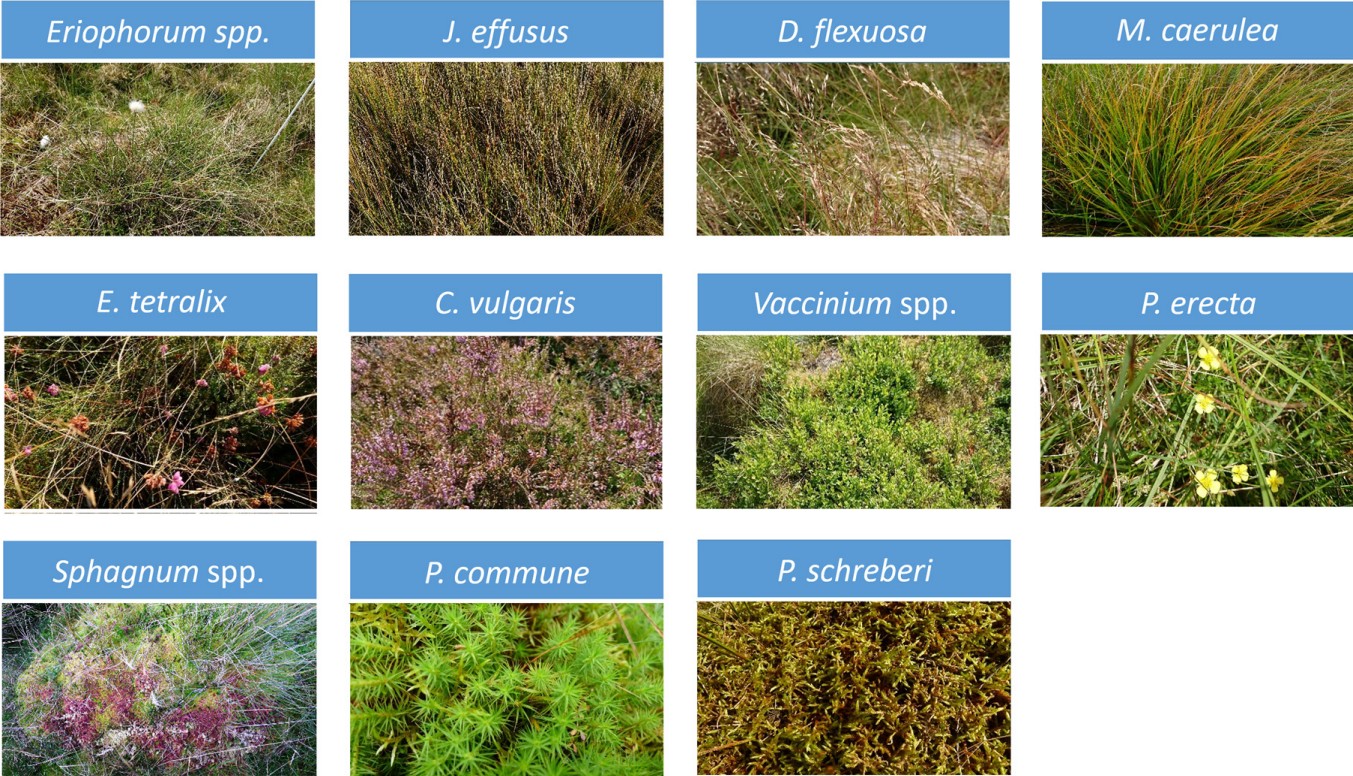

**Figure 3.** Photographs of the eleven dominant upper-canopy species identified by the ground survey. Photo credit: G. Simpson.

**Table 2.** Size of the 'dominant species' regions of interest (ROIs) collected for training the classification models and validating the outputs. Shown are the total number of polygons surveyed for each ROI and the total number of pixels contained therein.

| ROI Class | Total No. Polygons | Total No. Pixels |
|---|---|---|
| *Eriophorum vaginatum* | 39 | 11,361 |
| *Juncus effusus* | 18 | 27,887 |
| *Deschampsia flexuosa* | 23 | 7118 |
| *Molinia caerulea* | 25 | 25,284 |
| *Erica tetralix* | 20 | 3442 |
| *Calluna vulgaris* | 19 | 8358 |
| *Vaccinium* spp. | 16 | 3524 |
| *Potentilla erecta* | 19 | 882 |
| *Sphagnum* spp. | 11 | 2157 |
| *Polytrichum commune* | 32 | 7264 |
| *Pleurozium schreberi* | 34 | 8666 |
| **SUM** | 256 | 105,943 |

*2.4. Vegetation Classification*

2.4.1. Pixel-Based Classification

The pixel-based classifications were conducted using the ENVI software (v5.5.3, Harris Geospatial Solutions, Inc., Boulder, CO, USA). This type of classification uses the concept of feature space to group pixels into classes based on the similarity of their spectral signature. As such, geospatial patterns in the image data are ignored, and pixels are assumed to be independent of each other. All the classifications conducted were supervised, in that the spectral properties of the training data (pre-defined classes) were used to classify each image pixel based on its proximity to the mean of the training classes. This study explored the performance of the Minimum Distance, Mahalanobis Distance and Maximum Likelihood (ML) algorithms. These were run using the collected GNSS points to extract training and validation data. The ENVI 'Spectral Angle Mapper' tool [51] was also used for image classification based on the (i) field-spectroscopy measurements from the ASD (temporally limited); and (ii) average spectra from the single-species ROIs in the imagery (not temporally limited). Hereby, the reference spectra for each class are assigned a given vector denoting their position in the n-*D* feature space (where n is the number of bands, *D* is the dimension). Pixels were then assigned to a given class based on their similarity to the reference spectra, defined by the user as an angle from the class-vector to each pixel-vector in the n-*D* space. Here, a smaller angle denotes a stronger similarity to the reference spectra and vice versa.

2.4.2. ROI Separability and Classification Accuracy

The spectral separability of the ROIs was examined using the Jeffries–Matusita distance and Transformed Divergence measures [52,53]. The accuracy of the classifications was assessed using a confusion matrix, whereby the classification output classes were compared against the validation ROIs. At the image level, the accuracy was reported using two metrics: (i) the overall accuracy (OA, i.e., the percentage of correctly classified pixels; [54]); and (ii) the Kappa coefficient, or 'Kappa' [55], which is a measure of agreement between the classification output and the ground validation ROIs. The values of the Kappa coefficient range from zero (i.e., no agreement) to one (i.e., perfect agreement). Finally, for the individual classes, the Producer- and User accuracy (PA, UA, respectively) are reported, where PA indicates how much of the validation data were classified correctly by

the classification algorithm (i.e., errors of omission), and UA indicates how often classified pixels were actually present in the ground validation data (i.e., errors of commission).

### 2.4.3. Examining the Impact of Spatial and Temporal Sampling

The impact of the spatial resolution and UAV survey date on the image classification was assessed, as outlined in Figure 4. Firstly, the influence of the spatial resolution on the image classification was examined for a single day during the peak growing season (3 August 2021). Here, the original imagery was used (2.8 cm ground sampling distance; GSD) and re-sampled to create three coarser datasets of 5.6 cm, 11.2 cm and 22.4 cm GSD using the nearest neighbour method. The classification outputs from these four spatial datasets were compared to quantify the impact of the spatial resolution on the classification accuracy.

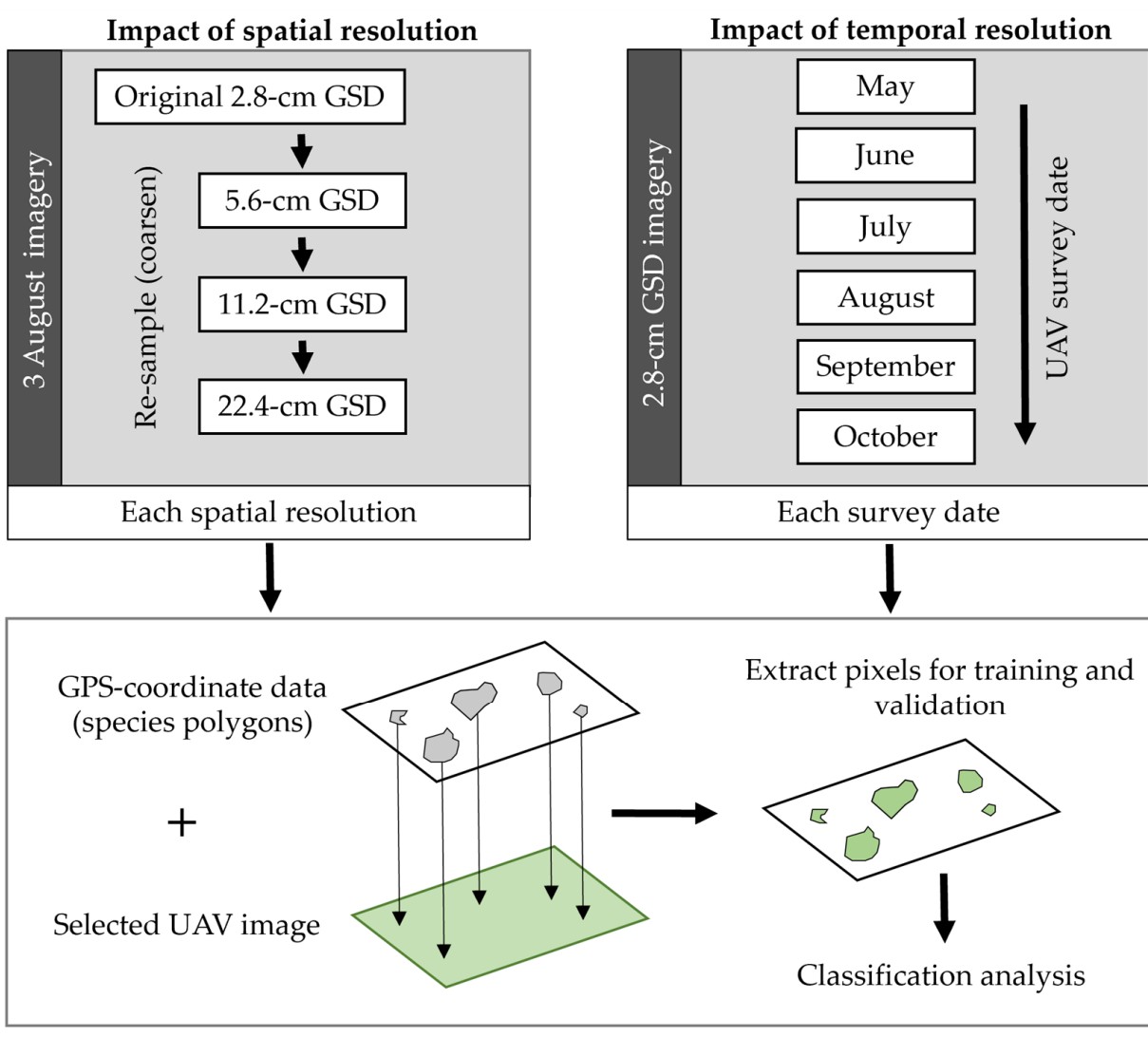

**Figure 4.** Outline of the data used to explore the impact of the spatial and temporal resolutions of the UAV imagery on the classification accuracy.

The impact of the sampling date on the vegetation classification was examined by conducting analyses on the original (2.8 cm GSD) UAV imagery collected on seven dates over the growing season (14 May–15 October 2021). Separate classifications were produced for each flight day, before the classification accuracy and the separability of the training data were compared for each species over the seven flight dates.

### 3. Results

*3.1. Processed UAV Imagery and Spectral Data*

A processed RGB and multispectral orthomosaic of the study area, flown at 65 and 25 m, respectively, are presented in Figure 5. The hummock–hollow micro-topography is evident from the imagery as well as the historic drainage ditches, which are identifiable as parallel line features running diagonally across the site. The differences between the two image sets highlight the strong ability of multispectral data to capture the spatial heterogeneity of vegetation at the site.

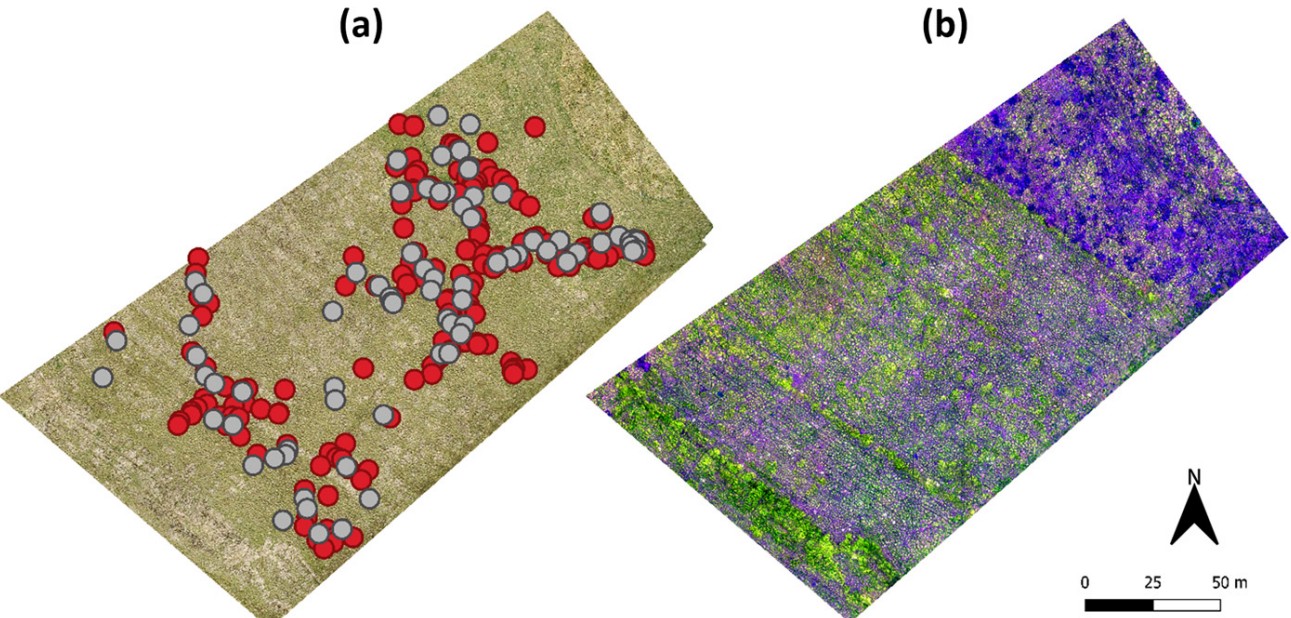

**Figure 5.** RGB (true-colour composite, (**a**)) and multispectral (false-colour composite, (**b**); RGB = b4-b2-b1) orthomosaics of the study area taken on 3 August 2021. The RGB image on the left (**a**) shows the location of training (red) and validation (grey) data points used in the classification analysis.

The spatial resolution of the generated multispectral reflectance maps (i.e., GSD) ranged from 2.71 to 2.93 cm and was influenced by the meteorological conditions (wind speed and direction) on each survey date. All the reflectance maps displayed a high accuracy for the geolocation, with an RMSE in the x–y direction of <1.5 cm (see Table 1). The processed RGB orthomosaic had a spatial resolution of 1.46 cm and again had a high geolocation accuracy (RMSE <1.2 cm in the x- and y-directions). The DSM had a vertical accuracy of 4 cm. Image analysis found that around 14% of the scene consisted of shaded areas, defined by a manufactured $GR_{av}$ band threshold. Finally, the normalised DSM and multispectral image stack were combined to create a 5-band image used in all the subsequent classification analyses. Note that because of the differences in the GSD of the two datasets, the normalised DSM was resampled to match the spatial resolution of the multispectral Parrot Sequoia imagery.

The field spectral reflectance measurements taken of vegetation at the site displayed considerable overlap between the dominant species and are presented in Figure 6. These measurements show the unique spectral signature and strong reflectance observed for *Sphagnum* at Auchencorth Moss. Figure 6 also highlights the large amount of detail in the spectral signature that is lost when extracting only the four bands sampled by the Parrot Sequoia.

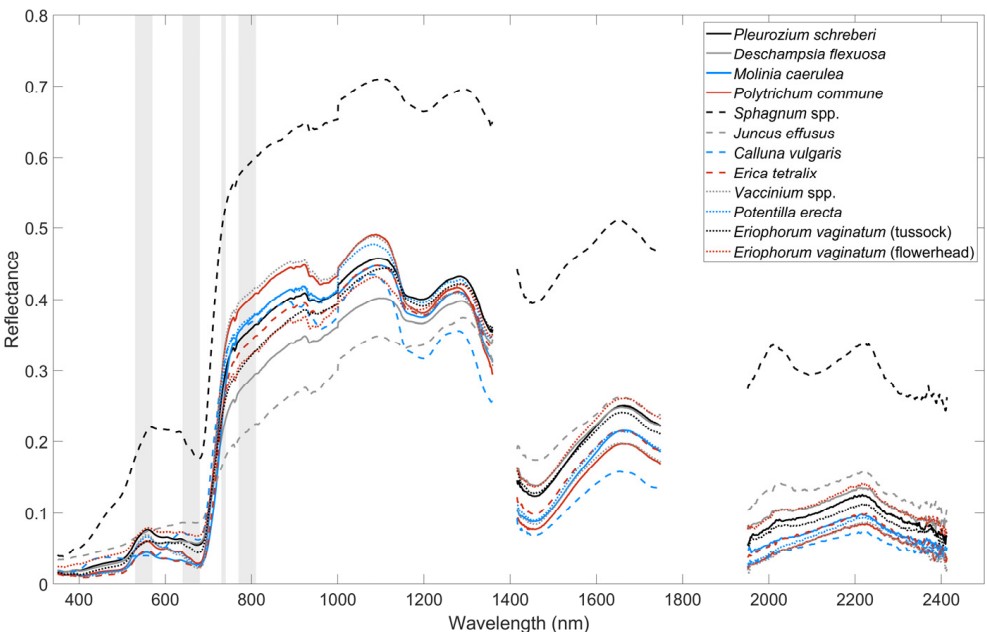

**Figure 6.** Spectral signatures of the dominant vegetation species at Auchencorth Moss. The reflectance data shown are from the field spectroscopy measurements conducted with the ASD on 23 July 2021. The gaps in the presented spectra result from the removal of bands affected by noise and atmospheric water absorption features. The wavelengths sampled by the four Parrot Sequoia bands (green, red, red-edge, and NIR, respectively) are depicted by the vertical grey bars towards the left-hand side of the plot.

### 3.2. Classification Analysis and Accuracy

The supervised classification methods examined in the present study varied widely in their accuracy when mapping the dominant species at Auchencorth Moss. Using original imagery from the peak growing season (3 August 2021), the overall accuracy (OA) ranged from as low as 18% to around 70%. On the whole, the ML (OA = 69%, Kappa = 0.63) and Mahalanobis Distance (OA = 60%, Kappa = 0.54) classifiers produced the most accurate classification maps. In contrast, the Minimum Distance classifier performed poorly (OA = 47%, Kappa = 0.39), as did the spectral angle classifier run using the GNSS coordinate training ROIs (OA = 28%, Kappa = 0.21). Finally, although not directly comparable, considerably poorer results were obtained when running the Spectral Angle Mapper with the resampled field spectroscopy data from 23 July (OA = 18%).

A confusion matrix was used to explore the output from the best-performing ML classifier in more detail (Table 3). The species *C. vulgaris*, *J. effusus* and *Sphagnum* were classified with the highest accuracy, and all three were associated with a high User- and Producer accuracy (UA, PA) of around 80–90%. The ML classifier also achieved high accuracy in mapping the shrub species *Vaccinium*, as well as the moss *P. schreberi*. In contrast, the grass species *D. flexuosa* and *M. caerulea* were classified with lower success (PA~60%). The confusion matrix shows that a large proportion of pixels were incorrectly classified as *D. flexuosa* instead of *M. caerulea* and *P. schreberi* (UA = 44%). Interestingly, although the ML classifier had a relatively high User accuracy for the class *E. vaginatum*, it had a low Producer accuracy (PA = 32%). In fact, pixels known to contain this species were more often classified as *D. flexuosa* or *M. caerulea*. The classification accuracy of *P. commune* was also poor, and this species was often misclassified as *J. effusus* (PA = 46%). Finally, the lowest accuracies were observed for the class with the smallest training and validation datasets, *P. erecta* (see Table 2). This forb was often misclassified as *D. flexuosa* or *P. commune* and was the class with the lowest UA of only 6%.

**Table 3.** Confusion matrix based on a Maximum Likelihood (ML) classification of the peak growing season imagery (3 August 2021). The analysis was conducted using original resolution data (2.8 cm GSD) in a 5-band image stack (multispectral and normalised DSM data), and GNSS survey point data were employed for the ground validation. Columns represent ground validation data, and rows represent classification output based on the training data. Note, values in the table's centre denote the number of pixels, with those in bold along the diagonal indicating the number of pixels for which the classification predicted the same species as in the ground validation data. The User and Producer accuracy are given as percentage values for ease of reference, with overall classification accuracy indicated in bold.

| | | Ground Validation Data | | | | | | | | | | | |
|---|---|---|---|---|---|---|---|---|---|---|---|---|---|
| | | *P. schreberi* | *D. flexuosa* | *M. caerulea* | *P. commune* | *Sphagnum* spp. | *J. effusus* | *C. vulgaris* | *E. tetralix* | *Vaccinium* spp. | *P. erecta* | *E. vaginatum* | **User Acc.** |
| **Classification output** | *P. schreberi* | **1544** | 54 | 26 | 31 | 0 | 258 | 7 | 23 | 0 | 1 | 267 | 70% |
| | *D. flexuosa* | 153 | **1181** | 240 | 107 | 0 | 1 | 38 | 154 | 1 | 54 | 728 | 45% |
| | *M. caerulea* | 0 | 0 | **1023** | 0 | 0 | 0 | 0 | 0 | 50 | 0 | 606 | 61% |
| | *P. commune* | 12 | 196 | 71 | **684** | 3 | 190 | 53 | 17 | 0 | 43 | 13 | 53% |
| | *Sphagnum* spp. | 54 | 1 | 0 | 0 | **462** | 0 | 0 | 0 | 0 | 0 | 8 | 88% |
| | *J. effusus* | 196 | 64 | 0 | 421 | 0 | **6412** | 0 | 0 | 7 | 23 | 63 | 89% |
| | *C. vulgaris* | 0 | 0 | 19 | 35 | 0 | 0 | **1202** | 17 | 21 | 0 | 0 | 93% |
| | *E. tetralix* | 10 | 15 | 16 | 0 | 0 | 0 | 195 | **687** | 0 | 0 | 38 | 72% |
| | *Vaccinium* spp. | 0 | 0 | 23 | 13 | 0 | 0 | 0 | 0 | **897** | 26 | 235 | 75% |
| | *P. erecta* | 114 | 361 | 70 | 160 | 9 | 174 | 34 | 23 | 0 | **85** | 358 | 6% |
| | *E. vaginatum* | 130 | 31 | 206 | 38 | 48 | 33 | 9 | 2 | 14 | 29 | **1091** | 67% |
| | **Producer Acc.** | 70% | 62% | 60% | 46% | 89% | 91% | 78% | 74% | 91% | 33% | 32% | **69%** |

*3.3. Impact of Spatial Resolution*

The original 2.8 cm GSD imagery was resampled to three coarser resolutions (5.6 cm, 11.2 cm and 22.4 cm GSD) to examine the impact of the spatial resolution on the imagery itself and the subsequent vegetation classification. A visual comparison is presented in Figure 7 and shows the large amount of detail captured in the highest resolution multispectral imagery (2.8 cm GSD), including the structure of individual grass tufts. This detail, although less sharp, was also present in the mid-resolution imagery (5.6 cm and 11.2 cm GSD). In contrast, the imagery with the coarsest resolution (22.4 cm GSD) appeared pixelated and individual vegetation features were no longer discernible. It follows that the classifications produced using higher-resolution imagery were more spatially heterogeneous (Figure 7). The ML classification employing the 2.8 cm GSD imagery, for example, produced classes that varied over short distances (<1 m), whereas in the classification produced using the coarsest imagery, the distances between class boundaries commonly spanned >2 m.

The accuracy of the classifications produced with this resampled imagery are provided in Table 4. Overall, the classification accuracy was found to increase with the increasing spatial resolution of the input data. Interestingly, although the highest accuracy was achieved using the original 2.8 cm GSD imagery (OA = 69%, Kappa = 0.63), the accuracy of the two medium-resolution classifications was only marginally lower. In contrast, a marked drop in classification accuracy was observed when coarsening the resolution of the input imagery from the 11.2 to 22.4 cm GSD (>22 percentage points for OA; 40% reduction in Kappa). Despite the low overall accuracy achieved using this coarse imagery, the success at the species level was variable. For example, pixels classified as *J. effusus*, *Vaccinium-*

and *Sphagnum* all had PA > 70%, whereas the PA for all the other classes was below 40%. Finally, the classification accuracy for species with a smaller patch size exhibited the largest reductions in accuracy when coarsening the input imagery from 11.2 cm to 22.4 cm GSD. The producer accuracy for *E. tetralix*, for example, dropped from 59% to 0%.

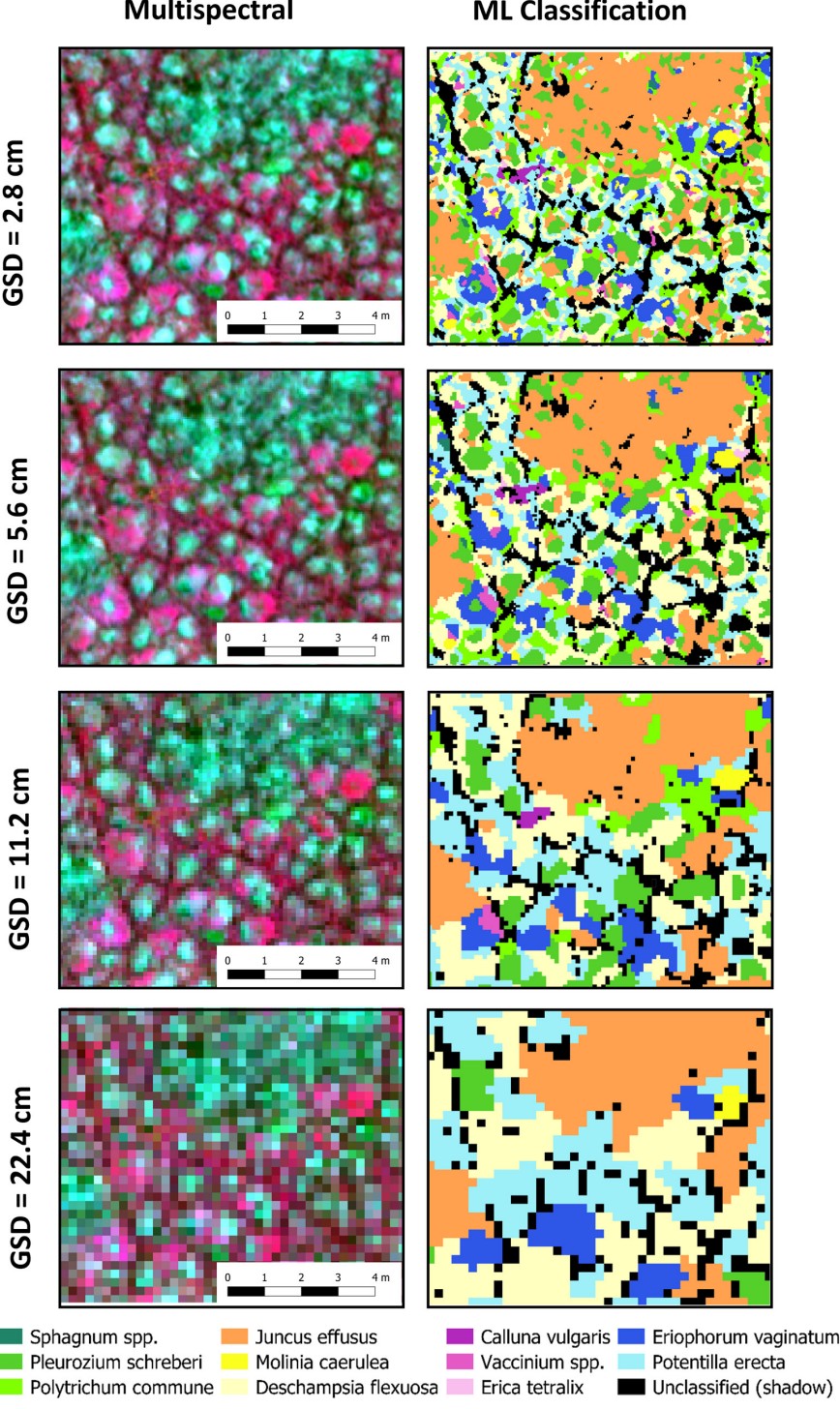

**Figure 7.** Impact of the spatial resolution on the image classification. Shown are false-colour composites of the multispectral imagery (left; RGB = b4-b2-b1) used as input for the Maximum Likelihood (ML) classification (right; see legend). A coarsening image resolution is shown from top to bottom (2.8 cm GSD to 22.4 cm GSD). Note, these classifications were conducted using the GNSS survey point data for ground validation.

**Table 4.** Impact of the spatial resolution on the classification accuracy. Shown are accuracy statistics for a Maximum Likelihood (ML) classifier run with imagery collected on 3 August 2021, employing GNSS point data as ground validation. The original 2.8 cm GSD dataset was resampled to produce three sets of coarser resolution imagery used in the classification.

| Spatial Resolution | Classification Accuracy | |
| --- | --- | --- |
| | Overall Accuracy | Kappa Coefficient |
| 2.8 cm GSD | 68.5% | 0.63 |
| 5.6 cm GSD | 65.4% | 0.60 |
| 11.2 cm GSD | 64.9% | 0.59 |
| 22.4 cm GSD | 42.8% | 0.35 |

### 3.4. Impact of Temporal Sampling

Changes in the illumination geometry and phenology were observed over the duration of the study, which spanned the greening, flowering, and senescence of vegetation. This seasonal variability was evident in the monthly UAV imagery, with a visible contrast in the spectral reflectance of vegetation between surveys (see Figure 8). Differences in the location and areal extent of shadow (defined by our manufactured $GR_{av}$ band) were also visible between survey dates (black shaded areas, Figure 9), meaning there was some variability in the image pixels employed for the classification analysis.

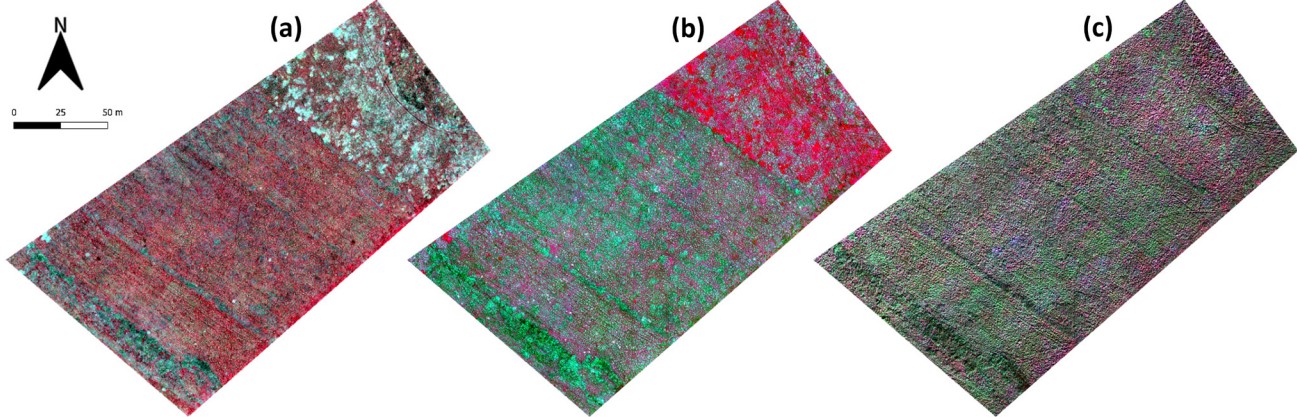

**Figure 8.** Multispectral imagery of Auchencorth Moss over the 2021 growing season. Shown are false-colour composites (RGB = b4-b2-b1) of the survey area acquired on three dates: 14 May (**a**); 3 August (**b**); and 15 October (**c**).

As seen for the spatial resolution, the ML classifier consistently produced the most accurate classifications across all the dates. The highest overall accuracy was achieved using imagery collected in late July and early August, when the growing season was at its peak (OA = 65.4%, Kappa = 0.63; OA = 68.5%, Kappa = 0.59, for 23 July and 3 August, respectively). In contrast, the least accurate classification was produced using imagery collected at the end of the growing season (15 October, OA = 38%, Kappa = 0.31), when senescence had already occurred. Despite these contrasting examples occurring at opposite stages of the growing season, no consistent trend in the overall accuracy over time was observed. The classification using imagery from 14 May, for example, was the third most accurate of all the surveyed dates (OA = 65%, Kappa = 0.58). This relatively high accuracy was achieved despite the minimal new growth at this point in the growing season.

The analysis highlighted some clear differences in the vegetation maps produced over the growing season, a selection of which are shown in Figure 9. During the early growing season, for example, a large portion of the survey area was classified as the graminoid *M. caerulea*. However, the predicted cover of this species was substantially lower in the

classifications employing peak and late growing season imagery. Similarly, the classification of pixels as the moss species *P. commune* was widespread using imagery from the peak growing season (Figure 9b), whereas visibly lower coverage was estimated using imagery from other dates.

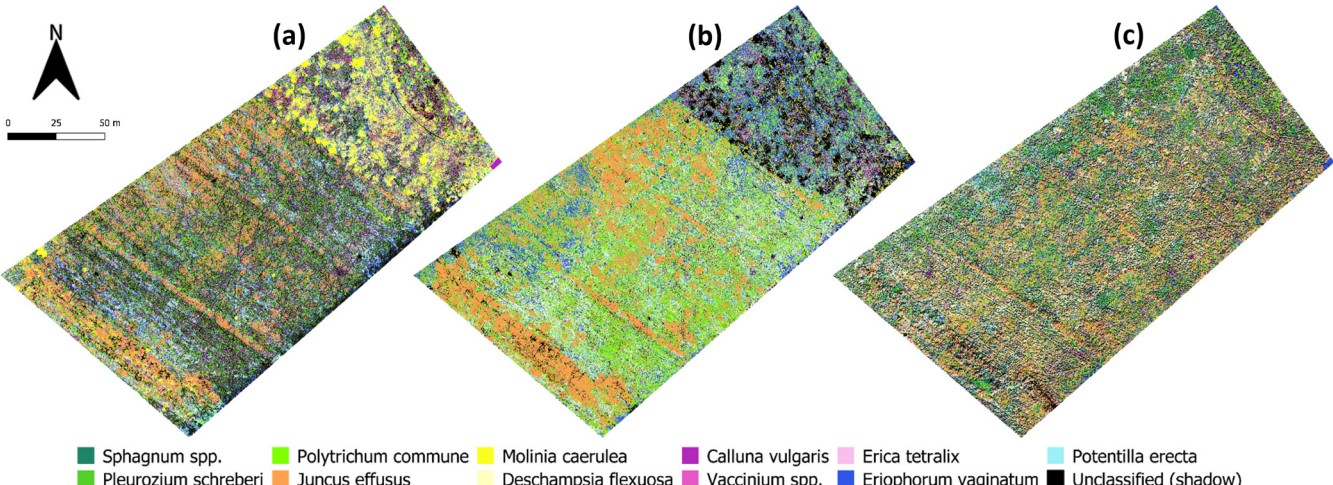

**Figure 9.** Temporal vegetation classifications of Auchencorth Moss over the 2021 growing season. Shown are the results of the Maximum Likelihood (ML) classifier run with imagery acquired on 14 May (**a**); 3 August (**b**); and 15 October (**c**). Note, the presented classifications varied in accuracy and were conducted using the GNSS survey point data for ground validation.

At the species level, the classification accuracy exhibited seasonal variability over the course of the phenological cycle. For example, examining data from the training ROIs showed that *Sphagnum* was spectrally distinct in the sampled sequoia bands for the majority of the growing season. During the early growing season (May–June), *Sphagnum* pixels from the training data exhibited higher reflectance in the red-edge and NIR bands than all the other species classes. Then, during the peak growing season (July–August), *Sphagnum* pixels exhibited an increased reflectance in the green and (to a lesser extent) red bands. It follows that the classification accuracy for this species was highest during these time periods (PA, UA $\geq$ 80%). The rush *J. effusus* also appeared distinctive in the red-edge and NIR bands, where the pixels had substantially lower reflectance values than the other classes. This species was most separable during the peak growing season (July–August) as, unlike the other classes, it did not exhibit a reflectance peak in the red-edge and NIR bands. Other distinctive species in the present study were *P. schreberi*, which demonstrated a high reflectance in the green band at the end of the growing season, and *M. caerulea*, which had by far the highest reflectance in the green and red bands of all the classes at the start of the growing season. These differences highlight the strong impact of vegetation phenology on the classification accuracy at Auchencorth Moss.

## 4. Discussion

### 4.1. Classification Analysis

#### 4.1.1. Classification Accuracy

This study used ultra-high-resolution UAV data to map vegetation at the species level in a temperate peatland. The impact of (i) the chosen supervised classification algorithm; (ii) the spatial resolution of the imagery (GSD); and (iii) the temporal sampling date on the classification accuracy was examined. The ML algorithm was found to produce the most accurate classification results at Auchencorth Moss for all the spatial scales and survey dates examined. Running this classifier with GNSS point data for ground validation, it was possible to achieve a relatively high accuracy at the species level (OA = 69%). In contrast,

the lowest accuracies (OA < 30%) were achieved using a spectral angle classifier run with GNSS point data or field spectroscopy measurements as validation data.

A number of studies in the existing literature report a higher classification accuracy than that presented for Auchencorth Moss. Kalacska et al., for example, classified *E. vaginatum* with 96% overall accuracy using RGB imagery (4.5 cm GSD) in a pixel-based supervised classification [32]. A similarly high accuracy (92%) was achieved by Lehmann et al., who classified vegetation at the species level in a Patagonian bog [56]. Despite the high accuracies reported above, it is important to note that the studies vary considerably in terms of their design (e.g., spatial resolution, classification methods, and vegetation groupings). These differences make it difficult to compare the results from the literature. In the Patagonian study, for example, the micro-topography was less pronounced than at Auchencorth Moss, and the authors employed a number of broad groupings, such as 'pools', 'dead vegetation' and 'lichens', in addition to their species-level classes [56]. This inclusion of strongly contrasting classes may have led to the high accuracy observed.

The present study was unique in that it mapped all 11 dominant species found in a site with strong spatial heterogeneity. Furthermore, many of the dominant species at Auchencorth Moss were spectrally similar (see Figure 6) and not clearly distinguishable from visual observation of the collected imagery alone (Figure 3). In contrast, the community types commonly mapped in the literature are often visually distinct in aerial imagery (e.g., [27]). This strong separability of classes likely leads to higher classification accuracy in these cases.

### 4.1.2. Choice of Methodology

The choice of classification methodology employed in the present study is important to consider when discussing the results, as it can have a large impact on the classification accuracy. We explored pixel-based classifications, whereas a number of studies in the literature employed an object-based approach. Object-based image analysis (OBIA) works on the basis that neighbouring pixels in very-high-resolution imagery often belong to the same class. This fact is ignored in the pixel-based approach we employed, which assumes individual pixels are independent of each other. OBIA segments the image into clusters of neighbouring pixels (i.e., 'objects'), before classifying each segment based on the training data. This approach is thought to lead to better results for very-high-resolution imagery [57,58]. It would be of interest in future research to examine how this approach performs with the patchy mix of vegetation at Auchencorth Moss.

Secondly, the present study was limited in that it explored only traditional classification algorithms (e.g., Maximum Likelihood, Minimum Distance). Although these algorithms achieved a high classification accuracy, another approach receiving a lot of attention in the literature is geospatial artificial intelligence (GeoAI; [59–61]). GeoAI has been found to achieve high accuracies in a number of studies (e.g., [46,59,62]) and is a powerful tool for exploring large datasets. This field is rapidly expanding and encompasses both machine-learning (e.g., random forest, support vector machines) and deep-learning algorithms (e.g., convolutional neural networks). That said, in comparison to traditional algorithms, GeoAI is a novel and developing field [59]. AI methods are often less accessible than traditional techniques, particularly for UAV-based studies (although frameworks are starting to be developed, e.g., [63]). At present, there is a lack of published literature comparing the two approaches for species-level classifications in peatlands, or even similar (e.g., wetland, grassland) ecosystems. We note, however, that machine-learning studies do not always report accuracies higher than we achieved at Auchencorth Moss using a Maximum Likelihood classifier (e.g., [22,41,62,64,65]). While a comparison of the performance of the two approaches was outside the scope of the present study, such work would be of great value in guiding the design of future classification studies.

### 4.1.3. Misclassification of Species

The misclassification of individual species at Auchencorth Moss likely occurred for multiple reasons. Firstly, species within the same PFT are known to be spectrally similar [66]. Hence, it can be difficult to separate vegetation within a given PFT at the individual-species level [65]. At Auchencorth Moss, the sedge *E. vaginatum* and the grasses *D. flexuosa* and *M. caerulea* are all tussock-forming species, and they were observed to have similar spectral signatures (see Figures 3 and 6). This often led to misclassification. Pixels known to contain *E. vaginatum*, for example, were more frequently misclassified as *D. flexuosa* or *M. caerulea* than classified correctly. In contrast, the high classification accuracy observed for moss species such as *Sphagnum* reflects their unique spectral signature, especially when compared to vascular plant species (Figure 6, see also [67,68]). Secondly, it is acknowledged that a large amount of spectral information was lost by limiting the sampling to the four Parrot Sequoia bands, and this likely led to the low accuracies achieved with the spectral angle mapper. In this respect, upgrading the sensor would provide more data for classification. Intermediate upgrades from the 4-band Parrot Sequoia sensor we utilised include the 5-band MicaSense (RedEdge-Mx, AgEagle Aerial Systems Inc., Wichita, KS, USA), the 6-band Tetracam (Tetracam Inc., Chatsworth, CA, USA), and the 10-band MicaSense Dual Camera System. At the other end of the scale, upgrading to a hyperspectral sensor would substantially increase the spectral resolution of the data, although this would substantially increase both the capital costs (e.g., ~GBP 4000 for our multispectral system vs. ~GBP 300,000 for a hyperspectral system) and the amount of processing required. An alternative would be to employ a larger range of data types for the classification analysis. Studies have shown, for example, that the use of multiple sensors (e.g., RGB, thermal, multispectral) can aid in the identification of species with strong spectral similarity [46]. A simpler approach using the collected data would be to employ vegetation indices or texture metrics alongside single-band reflectance to provide additional information for vegetation classification.

The multi-layer canopy of vegetation at Auchencorth Moss and the lack of spectrally 'pure' (i.e., 100% single-species) pixels will have introduced error into the present study. For example, patches of the moss *P. commune* were observed within the *J. effusus* canopy during the ground survey, and subsequently the analysis found that *P. commune* was often misclassified as *J. effusus*. The tall, thin *J. effusus* stems surrounding these moss patches, in combination with the viewing angles used in SfM, are likely to have caused *J. effusus* to be the dominant spectral signature from these pixels. Finally, it is noted that some species included for classification in our study had a limited number of training and validation ROIs (e.g., the forb *P. erecta*). These species were characterised by a small patch size and less extensive coverage, making it difficult to obtain sufficient sample points (see Table 2). The resulting 'class imbalance' will have impacted the produced classifications, with low class accuracy being linked to under-represented classes [54,69]. While a randomly distributed sampling strategy would in theory have been ideal and less affected by bias, at Auchencorth Moss the variable areal extent of the chosen classes and time constraints concerning field data collection mean that this methodology would be unlikely to capture the full variability in vegetation.

### 4.2. Impact of Spatial Resolution

Remote-sensing studies collecting imagery at higher spatial resolution provide more detailed datasets for classification. Such studies allow for data to be coarsened in order to (i) identify the best resolution for the intended mapping purpose; and (ii) minimise the processing time for the classification. In the present study, the classification accuracy improved when using imagery with higher spatial resolution (see Table 4). Using UAV imagery with <12 cm resolution, vegetation could be mapped at the species level with relatively high accuracy (i.e., OA ≥ 65%), despite the strong heterogeneity that characterised the site. In contrast, the coarsest dataset examined (22.4 cm GSD) lacked the resolution to capture vegetation species with smaller patch sizes (e.g., *E. tetralix*). These findings highlight

the need to consider the physical distances characterising the vegetation heterogeneity (e.g., patch size) at sites prior to surveying in order to produce the most accurate maps.

Despite its allure, obtaining and working with ultra-high-resolution data can be problematic. During UAV surveys, for example, wind and propwash (particularly if flying at low elevations, e.g., <5 m above ground level) can cause the vegetation itself (i.e., individual stems, flowers) to move between image acquisitions. This movement can lead to problems with geo-referencing and the matching of key points, and it can affect the quality of the output [70]. The GSD also has a strong influence on the signal-to-noise ratio. Vegetation appears more heterogeneous when imaged at spatial resolutions higher than that of the individual plants, as is often the case with centimetre-resolution imagery. In this case, rather than a single pixel covering the entire plant, the pixels will capture different parts of its structure (e.g., with separate pixels covering the leaf, stem, flower) and cause a higher contrast in the spectral signature [71]. This increased heterogeneity at finer scales can cause problems for classification referred to as the 'salt-and-pepper' effect, reducing the classification accuracy due to the high amounts of noise in the imagery. This noise particularly impacts pixel-based classifications, as were conducted in the present study [72,73], although its impact can be reduced by resampling the data. An alternative would be to employ object-based image classifications, which are less affected by noise as they segment image pixels into homogeneous objects [58].

Finally, the presence of shadow in high-resolution imagery can be pronounced, affecting the spectral reflectance measured by sensors and reducing the accuracy of subsequent classifications [37,38]. The contrast between shaded and illuminated pixels is particularly pronounced on sunny days, and various methods for masking shadow have been employed in the literature (e.g., [37,38,74]). The present study adopted a threshold approach in a manufactured $GR_{av}$ band (see Section 2.2.1), which categorised around 14% of image pixels as shadow. This thresholding approach likely overestimated the shadow's extent on overcast days (e.g., 3 August 2021; Figure 9b) compared to sunny days (e.g., 15 October 2021; Figure 9c). Although these differences may affect the classification accuracy, they are hard to evaluate, particularly due to difficulties quantifying the shadow fraction in overcast conditions. Nevertheless, the ultra-high-resolution imagery collected at Auchencorth Moss was used to create vegetation classifications of high accuracy, and it highlights the complexity of the studied ecosystem, with its hummock–hollow micro-topography and heterogeneous vegetation cover.

### 4.3. Impact of Temporal Sampling

In the present study, imagery acquired in the peak and very early growing season (late July–early August, and start of May) produced the most accurate vegetation classifications (OA ≥ 65%) for Auchencorth Moss. During the peak growing season, the vegetation was at its greenest and many species exhibited a more unique spectral signature. This increased level of separability between classes improved the classification accuracy. Examples include the pink–purple flower heads of *E. tetralix* and *C. vulgaris*, and the golden-brown seed heads of *J. effusus*. Interestingly, the analysis identified that *M. caerulea* was best classified using early-growing season imagery (PA = 81%, UA = 98%). This was the only deciduous grass species at the site and was widespread, creating large clumps of dead material at the end of the growing season [75]. Senescent vegetation is known to have a distinct spectral signature [39]. Hence, by the following year, this material exhibited a distinct peak in reflectance in the red band, which was not present for the other (non-deciduous) species. These findings highlight the strong impact of phenology on vegetation classification.

The optimal time for image acquisition will vary between sites, depending on the climate, environmental conditions, and the vegetation species present. However, the findings of the present study broadly coincide with those of Cole et al., who found that peatland PFTs in the Peak District (UK) were most separable in April and July, based on a time-series of hyperspectral field spectroscopy data [39]. It is therefore important that studies either (1) have detailed knowledge of the site phenology, limiting UAV surveys to the peak grow-

ing season; or (2) are able to survey on multiple dates (ideally during the early and peak growing season) to improve the classification accuracy. It is also worth noting that the temporal variability in spectral reflectance results can also be determined by environmental conditions (i.e., physiological status of the vegetation, see [76,77]). This is especially important to consider in peatlands, which exhibit strong heterogeneity at the microsite level. During dry periods, for example, although hollows can remain relatively moist, supporting vegetation growth, hummocks may dry out and induce the premature senescence of vegetation. Such small-scale variability can cause contrasting spectral signatures to be present for specimens within a single species and complicate classification analysis.

While the majority of UAV studies in the literature conducted classifications based on mono-temporal datasets, the present study has highlighted the strong impact of phenology over the growing season. In the literature, the use of multi-temporal datasets has been associated with improved image classification accuracy. Dudley et al., for example, employed a multi-temporal spectral library to map scrubby vegetation at the species level over a coastal elevation gradient in California [78]. The authors found that this improved the classification accuracy for some species compared to single-date libraries. The use of composite high-resolution imagery has not been widely explored for peatland vegetation mapping. This data gap may be due to the inaccessibility of many peatland environments and the lack of theoretically ideal meteorological conditions for spectral measurements (see [39]). However, due to the strong seasonal phenological cycle of peatland vegetation, such data would be valuable to collect, particularly if the use of UAVs for peatland study continues to increase in popularity.

## 5. Conclusions

UAV-based vegetation mapping can provide important data for peatland studies evaluating restoration projects (e.g., [25]) or estimating carbon emissions, particularly where certain vegetation communities, micro-topographical units, or species are associated with elevated fluxes of carbon dioxide or methane, for example [27,56]. Despite UAV technology allowing us to monitor peatlands at the centimetre level, to the best of our knowledge, this rich detail has rarely been exploited in vegetation classification.

The present study examined the use of ultra-high-resolution (2.8 cm GSD) multispectral UAV imagery for classifying vegetation at the species level. Despite strong spatial heterogeneity, 11 dominant species at the site were classified with overall accuracy of almost 70% using an ML classifier. The species *C. vulgaris*, *J. effusus*, and *Sphagnum* were classified with the highest accuracy, whereas the sedge *E. vaginatum* was often misclassified as the graminoid species *M. caerulea* or *D. flexuosa*. Surprisingly, little difference in accuracy for classifications employing imagery of 2.8 to 11.2 cm GSD was observed. However, a large drop in accuracy was seen when using imagery of coarser resolution (22.4 cm GSD). Further analysis and testing at a range of sites would help to better constrain the trends in accuracy under decreasing image resolution. This information would provide valuable guidance for future studies, allowing researchers to choose the most efficient study design.

The multi-temporal dataset analysed in the present study is unique. Analysis of this dataset has shown that the seasonal timing of image acquisition is a key factor influencing the classification accuracy. The most accurate vegetation classifications were produced using imagery acquired during the peak or early growing season, which supports previous findings [39]. In terms of the design, studies conducting more than one survey over the growing season would be most advantageous, particularly at sites where the phenology is not well known, whereas a targeted sampling design in the peak growing season could be employed at regularly monitored sites.

Despite the complexity of mapping peatland vegetation at fine spatial scales, data from the present study show it is possible to map vegetation at the species level with high accuracy. Such data products could be employed to estimate peatland carbon emissions, or by peatland restoration managers as a quick and repeatable method to monitor project success. Further research would help to identify which platforms (or combinations thereof)

produce the highest-accuracy classifications and which types of classifier (e.g., object- or pixel-based) and algorithms (traditional or GeoAI) are best suited to classifying peatland vegetation at the single-species level. This work would be invaluable in guiding future studies in how to achieve the highest classification accuracies.

**Author Contributions:** Conceptualisation—G.S. and C.J.N.; methodology—G.S., C.J.N. and T.W.; data collection—G.S., T.W., C.J.N., A.H. and S.G.-P.; formal analysis—G.S.; writing (original draft preparation)—G.S.; writing (review and editing)—all authors; visualisation—G.S.; supervision—C.J.N. and C.H.; project administration—C.J.N. and C.H.; funding acquisition—C.J.N. and C.H. All authors have read and agreed to the published version of the manuscript.

**Funding:** G.S. was supported by an NERC Doctoral Training Partnership grant (NE/L002558/1). Research and monitoring activities at Auchencorth Moss are supported by the Natural Environment Research Council as part of the CEH's UK-SCAPE programme (award number NE/R016429/1).

**Data Availability Statement:** The data presented in this study may be made available on request from the corresponding author. The data are not publicly available due to ongoing analysis for publication.

**Acknowledgments:** We thank Matthew Jones and the field team from UKCEH (Edinburgh) for managing access to Auchencorth Moss. We are grateful to the Airborne Research and Innovation Facility at the University of Edinburgh, who loaned the UAV equipment and provided access to the image processing software. We thank the NERC Field Spectroscopy Facility (University of Edinburgh) for supplying us with a number of calibrated reflectance panels and ground targets. Finally, we also thank the field assistants from the University of Edinburgh for providing ground support during the UAV data collection.

**Conflicts of Interest:** The authors declare no conflicts of interest. The funders had no role in the design of the study; in the collection, analyses, or interpretation of data; in the writing of the manuscript; or in the decision to publish the results.

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
