# Peer review of "Species-Level Classification of Peatland Vegetation Using Ultra-High-Resolution UAV Imagery"

_drones, doi:10.3390/drones8030097_

Round 1

Reviewer 1 Report

Comments and Suggestions for Authors

The work presented is innovative since, as it is commented, there is very little research on the mapping and analysis of peatlands. Moreover, it could be a very important topic for determining the impact of climate change. Therefore, both the topic and the use of technology are very well focused and of great interest.

Regarding corrections:

The use of uncrewed, although it is not a mistake, the use of unmanned is more widespread.

On the other hand, ultra high spatial resolution, I would use very high spatial resolution.

The following questions also arise:

1.- What is it that makes these sites so inaccessible to take data in the field?

2.-The use of information from UAVs is very useful, especially in areas that are difficult to access. It can even serve as ground truth.

However, for work where continuous or relatively frequent mapping of vegetation cover is required. Have you considered what would be the accuracy of classifying with satellite imagery?

3.- I also consider that nowadays the use of supervised classification models has had a great advanced, but this work does not explore any type of automatic supervised classification technic as Random Forest or SVM.

4.-Finally, I would have found it interesting to train supervised classification models using satellite images with information collected by UAVs.

Thank you very much for your research and effort.

Best regards

Author Response

Dear Reviewer,

Thank you very much for taking the time to review this manuscript. Please find our detailed response shown in red below, with the corresponding revisions shown in track changes in the re-submitted files.

----------------------------------------------------------------------------------

Point-by-point response to comments:

Yes

Can be improved

Must be improved

Not applicable

Does the introduction provide sufficient background and include all relevant references?

( )

(x)

( )

( )

Are all the cited references relevant to the research?

(x)

( )

( )

( )

Is the research design appropriate?

(x)

( )

( )

( )

Are the methods adequately described?

(x)

( )

( )

( )

Are the results clearly presented?

(x)

( )

( )

( )

Are the conclusions supported by the results?

(x)

( )

( )

( )

Our additional text in response to comments number 1 and 2 below has added further background in the introduction section of this manuscript, and which we believe now provides sufficient background for the reader.

Comments and Suggestions for Authors

The work presented is innovative since, as it is commented, there is very little research on the mapping and analysis of peatlands. Moreover, it could be a very important topic for determining the impact of climate change. Therefore, both the topic and the use of technology are very well focused and of great interest.

Regarding corrections:

The use of uncrewed, although it is not a mistake, the use of unmanned is more widespread.

We agree that the term ‘unmanned’ is more commonly used, but would like to promote the use of ‘uncrewed’ as a more gender-neutral term.

On the other hand, ultra high spatial resolution, I would use very high spatial resolution.

We find that both terms are used in the literature. From our observations, ‘very high spatial resolution’ is often employed to describe spatial resolutions <1-2 m (e.g. Räsänen et al., 2019; Schepaschenko et al., 2019). As the imagery employed in this study is at 2.8-cm resolution, we use the term ‘ultra-high’ to emphasise the fine, centimetre scale of imagery used. This terminology is employed in a number of studies both in the literature and cited here (e.g. Räsänen et al., 2020), We therefore are keen to keep this terminology for consistency with published works.

Supporting references:

Räsänen, A., Juutinen, S., Aurela, M. and Virtanen, T., 2019. Predicting aboveground biomass in Arctic landscapes using very high spatial resolution satellite imagery and field sampling. International Journal of Remote Sensing, 40(3), pp.1175-1199.

Räsänen, A.; Juutinen, S.; Tuittila, E.S.; Aurela, M.; Virtanen, T. Comparing Ultra-High Spatial Resolution Remote-Sensing Methods in Mapping Peatland Vegetation. J. Veg. Sci. 2019, 30, 1016–1026, doi:10.1111/JVS.12769

Schepaschenko, D., See, L., Lesiv, M., Bastin, J.F., Mollicone, D., Tsendbazar, N.E., Bastin, L., McCallum, I., Laso Bayas, J.C., Baklanov, A. and Perger, C., 2019. Recent advances in forest observation with visual interpretation of very high-resolution imagery. Surveys in Geophysics, 40, pp.839-862.

The following questions also arise:

1.- What is it that makes these sites so inaccessible to take data in the field?

Firstly, many peatlands are located in geographically remote areas. This, in combination with their pronounced microtopography and boggy nature means they can often be impassible and would require extensive boardwalks which are not always possible to build due to resourcing. These factors make it time-consuming and expensive to conduct manual surveys.

We have added additional text to this effect in the introduction (see lines 53-55)

2.-The use of information from UAVs is very useful, especially in areas that are difficult to access. It can even serve as ground truth. However, for work where continuous or relatively frequent mapping of vegetation cover is required. Have you considered what would be the accuracy of classifying with satellite imagery?

We have not considered satellite-based image classification in the present study. The frequent presence of cloud cover in boreal and temperate regions (where many peatlands are located) makes it difficult to conduct frequent vegetation mapping based on optical satellite data (e.g. Connolly, 2019; O’Leary et al. 2023). Additional text has been added to this effect in the introduction (see lines 61-63).

In terms of vegetation classification accuracy using satellite data, this would be largely influenced by the spatial resolution of data employed. While the majority of panchromatic satellite imagery is available at resolutions <1 m (e.g. WorldView, QuickBird), multispectral datasets often provide imagery >1 m resolution (e.g. RapidEye, PlanetScope, Sentinel-2), which would mask a large amount of the spatial heterogeneity observed in the present study. As such, satellite-based studies often focus on mapping plant functional types, or and would not be comparable with the collected UAV data in the present study, which is the main focus of our research.

Supporting references:

Connolly, J., 2018. Mapping land use on Irish peatlands using medium resolution satellite imagery. Irish Geography, 51(2), pp.187-204.

O'Leary, D., Brown, C., Healy, M.G., Regan, S. and Daly, E., 2023. Observations of intra-peatland variability using multiple spatially coincident remotely sensed data sources and machine learning. Geoderma, 430, p.116348.

3.- I also consider that nowadays the use of supervised classification models has had a great advanced, but this work does not explore any type of automatic supervised classification technic as Random Forest or SVM.

We chose to focus on traditional supervised classification algorithms only (and we have another study only looking to machine learning), and therefore here we do not examine accuracy from any machine or deep-learning classification algorithms. Supervised methods are highly accessible, and successful, and we believe the use of ML approaches is also of importance and something we are addressing in a future paper. The reviewer comments that our current work presented in the manuscript has high academic value in its current form.

We found a lack of published literature comparing the two classification approaches at the species-level in peatlands, or even similar (e.g. wetland, grassland) ecosystems. However, we note that machine learning studies do not always report accuracies higher than we found using a maximum likelihood classifier. For example, results from a study mapping tropical peatlands reported classification accuracy using random forest of around 70% (Sencaki et al. 2020), which is comparable to the species-level accuracy obtained in our study. In another study, DeLancey et al. (2019) used machine learning algorithms to classify non-treed boreal peatlands with accuracies of 57% and 71% for (open fens and bogs respectively). Furthermore, we note that a number studies conducting broad peatland classifications using machine learning approaches again report similar accuracies to what we achieved with the maximum likelihood classifier, e.g. Erudel et al. (2017) using a range of algorithms including random forest, support vector machines and regularised logistic regression; and similarly both Räsänen et al. (2019, 2020) and Ingle et al. (2023) using random forest.

We have restructured the first part of our discussion to address this comment, including a new subsection, 4.1.2 ‘Choice of methodology’ - see in particular lines 542-558 therein; and lines 723-725 in the conclusion. After these additions, we feel that the MS now highlights the potential of machine learning/GeoAI for vegetation mapping at the species-level, without taking away from the main focus of our study which was to examine the effect of spatial and temporal resolution on image classification.

Supporting references:

DeLancey, E.R., Kariyeva, J., Bried, J.T. and Hird, J.N., 2019. Large-scale probabilistic identification of boreal peatlands using Google Earth Engine, open-access satellite data, and machine learning. PLoS One, 14(6), p.e0218165.

Erudel, T., Fabre, S., Houet, T., Mazier, F. and Briottet, X., 2017. Criteria comparison for classifying peatland vegetation types using in situ hyperspectral measurements. Remote Sensing, 9(7), p.748.

Ingle, R., Habib, W., Connolly, J., McCorry, M., Barry, S. and Saunders, M., 2023. Upscaling methane fluxes from peatlands across a drainage gradient in Ireland using PlanetScope imagery and machine learning tools. Scientific Reports, 13(1), p.11997.

Räsänen, A., Aurela, M., Juutinen, S., Kumpula, T., Lohila, A., Penttilä, T. and Virtanen, T., 2020. Detecting northern peatland vegetation patterns at ultra‐high spatial resolution. Remote Sensing in Ecology and Conservation, 6(4), pp.457-471.

Räsänen, A., Juutinen, S., Tuittila, E.S., Aurela, M. and Virtanen, T., 2019. Comparing ultra‐high spatial resolution remote‐sensing methods in mapping peatland vegetation. Journal of Vegetation Science, 30(5), pp.1016-1026.

Sencaki, D.B., Prayogi, H., Arfah, S. and Pianto, T.A., 2020, June. Machine learning approach for peatland delineation using multi-sensor remote sensing data in Ogan Komering Ilir Regency. In IOP Conference Series: Earth and Environmental Science (Vol. 500, No. 1, p. 012005). IOP Publishing.

4.-Finally, I would have found it interesting to train supervised classification models using satellite images with information collected by UAVs.

Yes, we too have many ideas going forward with the data we have collected. This would be a very large piece of additional work, which we feel would be better incorporated in a further manuscript.

Reviewer 2 Report

Comments and Suggestions for Authors

This study is considered to have high academic value as it presents the possibility of precise plant mapping at the species level using UAV-based ultra-high-resolution multispectral imagery.

However, it is deemed necessary to consider additional methodologies to improve the accuracy of vegetation classification, such as the exploration of the potential use of techniques like GeoAI.

Author Response

Dear Reviewer,

Thank you very much for taking the time to review this manuscript. Please find our detailed response shown in red below, with the corresponding revisions shown in track changes in the re-submitted files.

----------------------------------------------------------------------------------

Point-by-point response to comments

Yes

Can be improved

Must be improved

Not applicable

Does the introduction provide sufficient background and include all relevant references?

(x)

( )

( )

( )

Are all the cited references relevant to the research?

(x)

( )

( )

( )

Is the research design appropriate?

(x)

( )

( )

( )

Are the methods adequately described?

(x)

( )

( )

( )

Are the results clearly presented?

(x)

( )

( )

( )

Are the conclusions supported by the results?

(x)

( )

( )

( )

Comments and Suggestions for Authors

This study is considered to have high academic value as it presents the possibility of precise plant mapping at the species level using UAV-based ultra-high-resolution multispectral imagery.

However, it is deemed necessary to consider additional methodologies to improve the accuracy of vegetation classification, such as the exploration of the potential use of techniques like GeoAI.

We acknowledge, although GeoAI is still in its infancy, that these techniques show much promise for future classification studies (DeLancey et al., 2019; Janowicz et al., 2020; Song et al., 2023). We chose to focus on traditional classification algorithms, which are highly accessible and we show to be successful. AI methods in comparison are often less accessible, particularly for UAV-based studies, although frameworks are starting to be developed for this purpose (e.g. Ballesteros et al., 2022).

While a comparison of performance between GeoAI and traditional classification methods with our collected data is outside the scope of the present study, forthcoming work can have an AI focus, and it would be of interest for us to further explore the collected data using machine learning algorithms at a range of resolutions.

We do believe the MS can benefit from specific discussion of GeoAI techniques. We have therefore included additional text in the discussion and conclusion sections: giving a brief overview of GeoAI, its performance in similar studies, and its potential to increase classification accuracy in future mapping work (see new discussion subsection 4.1.2 ‘Choice of methodology’, in particular lines 542-558; and lines 723-725 in the conclusion section). After these additions, we feel that the MS now highlights the potential of GeoAI for vegetation mapping at the species-level, without taking away from the main focus of our study which was to examine the effect of spatial and temporal resolution on image classification.

References:

Ballesteros, J.R., Sanchez-Torres, G. and Branch-Bedoya, J.W., 2022. A GIS pipeline to produce GeoAI datasets from drone overhead imagery. ISPRS International Journal of Geo-Information, 11(10), p.508.

DeLancey, E.R., Kariyeva, J., Bried, J.T. and Hird, J.N., 2019. Large-scale probabilistic identification of boreal peatlands using Google Earth Engine, open-access satellite data, and machine learning. PLoS One, 14(6), p.e0218165.

Janowicz, K., Gao, S., McKenzie, G., Hu, Y. and Bhaduri, B., 2020. GeoAI: spatially explicit artificial intelligence techniques for geographic knowledge discovery and beyond. International Journal of Geographical Information Science, 34(4), pp.625-636.

Song, Y., Kalacska, M., Gašparović, M., Yao, J. and Najibi, N., 2023. Advances in geocomputation and geospatial artificial intelligence (GeoAI) for mapping. International Journal of Applied Earth Observation and Geoinformation, p.103300.

Reviewer 3 Report

Comments and Suggestions for Authors

This paper is well written, with a clear setup, methods, and results. It is highly relevant to those interested in mapping heterogeneous wetlands vegetation with high-resolution drone imagery. With very minor edits, it will be ready for publication. My recommended changes are:

On page 2, line 55 – It is unusual to abbreviate remote sensing as “RS”. I would just leave it as remote sensing in the paper.

Pg. 2, lines 85-86 – I would say “…and poses a challenge for mapping if using single time frames”. If you use multiple dates of imagery (as you do), then exploiting phenological changes can help improve accuracy.

Pg. 3, line 120 – is that 2cm horizontal accuracy? Judging by your comments later in the paper, I think you mean horizontal here, and you should say so.

Pg. 6, Figure 2 – I would label the graphs with the dates right above or below each one, if the journal allows that. I found myself having to hand-label them to keep the precise dates straight.

Pg. 7, line 235 (and following paragraph) – A better name for this subsection would be “Vegetation training data” – it isn’t really about GNSS measurements, which better describes how the locations of the data were recorded, not the importance of these data.

Pg. 20, lines 532-536. There are options between the 4-band Sequoia and a very expensive hyperspectral system, such as the 5-band Micasense, 6-band Tetracam, and 10-band multispectral system created by using the Micasense RedEdge and Blue systems together. I would mention these as an alternative that may increase accuracy without having to go all the way to expensive hyperspectral systems.

Author Response

Dear Reviewer,

Thank you very much for taking the time to review this manuscript. Please find our detailed responses shown in red below, with the corresponding revisions shown in track changes in the re-submitted files.

----------------------------------------------------------------------------------

Point-by-point response to comments

Yes

Can be improved

Must be improved

Not applicable

Does the introduction provide sufficient background and include all relevant references?

(x)

( )

( )

( )

Are all the cited references relevant to the research?

(x)

( )

( )

( )

Is the research design appropriate?

(x)

( )

( )

( )

Are the methods adequately described?

(x)

( )

( )

( )

Are the results clearly presented?

(x)

( )

( )

( )

Are the conclusions supported by the results?

( )

( )

( )

( )

Comments and Suggestions for Authors

This paper is well written, with a clear setup, methods, and results. It is highly relevant to those interested in mapping heterogeneous wetlands vegetation with high-resolution drone imagery. With very minor edits, it will be ready for publication. My recommended changes are:

On page 2, line 55 – It is unusual to abbreviate remote sensing as “RS”. I would just leave it as remote sensing in the paper.

We agree with this suggestion and have removed this abbreviation from the text.

Pg. 2, lines 85-86 – I would say “…and poses a challenge for mapping if using single time frames”. If you use multiple dates of imagery (as you do), then exploiting phenological changes can help improve accuracy.

Again, we agree and have changed the text as suggested.

Pg. 3, line 120 – is that 2cm horizontal accuracy? Judging by your comments later in the paper, I think you mean horizontal here, and you should say so.

We agree that this change would add more clarity and have the text accordingly.

Pg. 6, Figure 2 – I would label the graphs with the dates right above or below each one, if the journal allows that. I found myself having to hand-label them to keep the precise dates straight.

We have changed Figure 2 accordingly, adding titles to each of the subplots (see pg. 7)

Pg. 7, line 235 (and following paragraph) – A better name for this subsection would be “Vegetation training data” – it isn’t really about GNSS measurements, which better describes how the locations of the data were recorded, not the importance of these data.

We find that the term “vegetation training data” may confuse the reader. This term would apply to both subsections 2.3.1 ‘Spectral reflectance measurements’ and 2.3.2 ‘GNSS measurements’. The spectral reflectance measurements described in 2.3.1 are employed specifically as training data to conduct classifications using the spectral angle mapper; whereas the GNSS measurements are employed not only for vegetation training data, but also for assessing the geolocation accuracy of the UAV imagery orthomosaics, via the use of GCPs and check-points. 

Pg. 20, lines 532-536. There are options between the 4-band Sequoia and a very expensive hyperspectral system, such as the 5-band Micasense, 6-band Tetracam, and 10-band multispectral system created by using the Micasense RedEdge and Blue systems together. I would mention these as an alternative that may increase accuracy without having to go all the way to expensive hyperspectral systems.

We agree with this suggestion, and have added additional text highlighting the availability of these cheaper systems (see lines 572-578)

Round 2

Reviewer 1 Report

Comments and Suggestions for Authors

Thank you for all the answers to the previous review. I look forward to further research with all the ideas than have emerged from this article.